



# Technical Note: Combining Quantile Forecasts and Predictive Distributions of Stream-flows

Konrad Bogner[1], Katharina Liechti[1], and Massimiliano Zappa[1]

[1]Swiss Federal Institute for Forest, Snow and Landscape Research WSL, Birmensdorf, Switzerland

*Correspondence to:* K. Bogner (konrad.bogner@wsl.ch)

**Abstract.** The enhanced availability of many different hydro-meteorological modelling and forecasting systems raises the issue of how to optimally combine this great deal of information. Especially the usage of deterministic and probabilistic forecasts with sometimes widely divergent predicted future stream-flow values makes it even more complicated for decision makers to sift out the relevant information. In this study multiple stream-flow forecast information will be aggregated based on

several different predictive distributions, resp. quantile forecasts. For this combination the Bayesian Model Averaging (BMA) approach, the Nonhomogeneous Gaussian Regression (NGR), also known as Ensemble Model Output Statistic (EMOS) model and a novel method called Beta transformed Linear Pooling (BLP) will be applied. By the help of the Quantile Score (QS) and the Continuous Ranked Probability Score (CRPS), the combination results for the Sihl river in Switzerland with about five years of forecast data will be compared and the differences between the raw and the optimally combined forecasts will be

highlighted. The results demonstrate the importance of applying proper forecast combination methods for decision makers in the field of flood and water resources management.

## 1 Introduction

The combination, or aggregation, of differing probability distributions into a single one could result in beneficial effects, since the differences between various forecast systems provide a better understanding of the uncertainty about the target quantities

and the aggregates may reflect more accurately the information. However, the biggest advantage of aggregation is that the forecaster is not forced to decide a priori which forecast system is the most reliable at the actual point of issuing a forecast, because the combination method will be optimized at each forecast run by taking into consideration the quality of the forecast from previous time steps. Thus, the data itself will automatically lead to the optimal decision incorporating all available information about the different deficiencies and strengths of the individual forecast systems.

In econometrics and related disciplines, the combination of forecasts has a long tradition starting with Bates and Granger (1969) suggesting the use of empirical weights derived from 'out of sample' forecast variances. An overview over the last forty years of forecast combination in the economic field can be found in Wallis (2011). Thompson (1977) was one of the first who outlined the advantages of forecast combinations in meteorology and Shamseldin et al. (1997) showed different methods of combining the output of different hydrological models. In Abrahart and See (2002) different combination methods for hydro-

logical forecast models are compared. Diks and Vrugt (2010) compare different model averaging approaches, showing that a



simple regression method could result in improvements comparable to more sophisticated methods.

In general the challenge of model combination is that, apart from the simple model averaging methodologies, different weights need to be assigned according to the quality of the forecast of the preceding days and periods. A frequently used method for model averaging and forecast combination is the method of Bayesian Model Averaging (BMA) introduced by Min and Zellner

(1993) and Raftery et al. (1997), where the weights are based on posterior model probabilities within a Bayesian framework. The BMA method has been applied in the field of ensemble forecast calibration (Raftery et al. (2005); Fraley et al. (2010)) and for flood forecasting purposes, e.g. in Ajami et al. (2007), Vrugt and Robinson (2007), Todini (2008) and Hemri et al. (2013). In Gneiting et al. (2005) and Gneiting et al. (2007) the term calibration is used to describe the statistical consistency between the distributional forecasts and the observations and is a joint property of the predictions and the events that materialise. A

state of the art calibration and bias correction method is the Non-homogeneous Gaussian Regression (NGR), also known as Ensemble Model Output Statistic (EMOS) technique of Gneiting et al. (2005). It fits a single parametric predictive probability density function (pdf) using summary statistics from the (multi-model) ensemble and corrects simultaneously for biases and dispersion errors. Also NGR has been applied many times successfully for calibrating and combining hydro-meteorological ensemble forecasts (see for example Hemri et al. (2014)).

The Beta transformed Linear Pooling (BLP) approach, which has been developed recently by Ranjan and Gneiting (2010) and Gneiting and Ranjan (2013) for combining predictive distributions, will be tested and compared with the NGR and the BMA in this study. To the author's knowledge the BLP and the associated estimation of weights, which assign relative importance to the individual predictive distributions, has not been applied to hydrological forecasts so far.

Before the combination methods are applied, the errors of the hydrological model are corrected in order to minimize the dif-

ference between the last available observation and the predictions at the time of initialization of the forecast. This process of error correction is later on called post-processing, since it starts after completing the hydrological simulations and predictions given meteorological observations or forecasts. Depending on the post-processing method, quantiles or pdf's for future stream-flows will be derived for each single forecast time-step. Whereas Quantile Regression (QR) methods (Koenker (2005)) and modifications of it will lead to predictions of quantiles, a predictive pdf can be derived for example by the recently developed

waveVARX method (Bogner and Pappenberger (2011)) directly. For more details of these post-processing methods the reader is referred to Bogner et al. (2016), whereas the objective of this paper will be the analysis of combination methods of forecasts. In the next section the three combination methods and the applied verification measures will be described. After the presentation of the data and the results, the outcome of the comparison will be discussed and summarized in the conclusions.

## 2 Methods

The most widely used form of aggregation is to take an unweighted average of probabilities, probability densities or distribution functions. Stone (1961) named this method the linear opinion pool. Lichtendahl et al. (2013) have examined averaging quantiles of continuous distributions given by multiple information sources rather than averaging probabilities. Both approaches of probability and quantile averaging have been applied in this paper for averaging the post-processed Ensemble Prediction





System (EPS) based stream-flow forecasts.

However, for the subsequent combination of the forecasts aggregation methods will be used, which allow to assign different weights to the different raw and post-processed forecasts.

## 2.1 Bayesian Model Averaging (BMA)

If the combination is calculated within a Bayesian Framework by using weights corresponding to the posterior model probabilities, it is usually referred to as BMA and follows from direct application of Bayes' theorem as explained in e.g. Min and Zellner (1993) and Raftery et al. (1997).

In Raftery et al. (2005) the statistical BMA model is extended to dynamical forecast models, where each forecast and/or ensemble member is represented by a probabilistic distribution for which a weight is assigned based on the past performance of

each individual forecast. These weights are used to combine all distributions into one single mixture distribution. Therefore the BMA predictive model of the quantity of interest $y$ is given by

$$p(y|k_1,\ldots,k_M) = \sum_{m=1}^{M} h_m g_m(y|k_m), \tag{1}$$

where $h_m$ is the posterior probability (i.e. weight) of forecast $k_m$ being the best forecast derived from its performance in the training period and the conditional pdf of $y$ on $k_m$, $g_m(y|k_m)$, given that $k_m$ is the best forecast in the ensemble with

$m = 1,\ldots,M$ members, resp. models.

## 2.2 Non-homogeneous Gaussian Regression (NGR)

Another possibility to address under-dispersion and forecast bias is the use of the Non-homogeneous Gaussian Regression (NGR) method, also known as Ensemble Model Output Statistics (EMOS) and is based on multiple linear regression for linear

variables, such as temperature or stream-flows, and logistic regression for binary variables, such as precipitation occurrence or freezing. More information about the MOS technique can be found for example in Glahn and Lowry (1972) and Wilks (1995). Its extension for ensembles is explained in Gneiting et al. (2005) and a brief summary of this method is given hereafter. Let $y$ denote again the variable of interest (e.g. stream flow) and let $k_1,\ldots,k_M$ be the corresponding forecast of the $M$ ensemble members or models. If $\mathcal{N}(\mu,\sigma^2)$ denotes a Gaussian density with mean $\mu$ and variance $\sigma^2$, the NGR predictive distribution is

given by

$$y|k_1,\ldots,k_M \sim \mathcal{N}(a_0 + a_1 k_1 + \cdots + a_M k_M, b_0 + b_1 s^2), \tag{2}$$
$$\text{where } s^2 = \frac{1}{M} \sum_{m=1}^{M} \left( k_m - \frac{1}{M} \sum_{m=1}^{M} k_m \right)^2.$$

Thus the predictive mean is equal to the regression estimates with coefficients $a_0,\ldots,a_m,b_0$, and $b_1$ and forms a bias-corrected weighted average of the different forecasts (ensemble members), whereas the predictive variance depends linearly



on the variance of the forecast models (ensemble members).

## 2.3 Beta transformed Linear Pool (BLP)

In Ranjan and Gneiting (2010) it has been stated that any non-trivially weighted average of distinct probability forecasts will
be uncalibrated and lack sharpness, even when the individual forecasts have been calibrated. Hence they suggested a composite
of the traditional linear pool with a beta transform. The aggregation method introduced by Ranjan and Gneiting (2010) and
Gneiting and Ranjan (2013) considers the Beta transformed Linear Pool (BLP) for a set of predictive cdfs $F_1, \ldots, F_M$ as

$$F(y) = B_{\alpha,\beta} \left( \sum_{m=1}^{M} \omega_m F_m(y) \right) \tag{3}$$

for $y \in R$, where $B_{\alpha,\beta}$ denotes the cdf of the standard Beta distribution with parameters $\alpha > 0$ and $\beta > 0$ and $\omega_1, \ldots, \omega_M$ being
nonnegative weights that sum to 1. The BLP density forecast for the component densities $f_i, \ldots, f_M$ then is

$$f(y) = \left( \sum_{m=1}^{M} \omega_m f_m(y) \right) b_{\alpha,\beta} \left( \sum_{m=1}^{M} \omega_m F_m(y) \right) \tag{4}$$

with parameters $\alpha > 0$ and $\beta > 0$ of the Beta density function $b_{\alpha,\beta}$. For $\alpha = \beta = 1$ the BLP corresponds to the traditional
linear opinion pool.

Thus $B_{\alpha,\beta}$ can be interpreted as a parametric calibration function for combining $F_1, \ldots, F_M$ with mixture weights $\omega \in \Delta_M$,
which assign relative importance to the individual predictive distributions. The parameters $\alpha > 0$ and $\beta > 0$ and the weights
$\omega_1, \ldots, \omega_M$ are estimated with the maximum likelihood method. The log likelihood function for the BLP model (4) is

$$
\begin{aligned}
\ell(\omega_1, \ldots, \omega_M; \alpha, \beta) &= \sum_{j=1}^{J} \log(f(y_j)) \\
&= \sum_{j=1}^{J} \log \left( \sum_{m=1}^{M} \omega_m f_{mj}(y_j) \right) + \sum_{j=1}^{J} \log \left( b_{\alpha,\beta} \left( \sum_{m=1}^{M} \omega_m F_{mj}(y_j) \right) \right) \\
&= \sum_{j=1}^{J} \log \left( \sum_{m=1}^{M} \omega_m f_{mj}(y_j) \right) \\
&\quad + \sum_{j=1}^{J} \left( (\alpha-1) \log \left( \sum_{m=1}^{M} \omega_m F_{mj}(y_j) \right) \right. \\
&\quad \left. + (\beta-1) \log \left( 1 - \sum_{m=1}^{M} \omega_m F_{mj}(y_j) \right) \right) + J \log B(\alpha, \beta)
\end{aligned}
\tag{5}
$$

where $B$ is the classical Beta function.

This BLP approach has been applied now to combine the different forecast systems.



## 2.4 Verification

Although probability and quantile forecasts are both probabilistic products, the former is expressed in terms of a probability (e.g. that a certain threshold will be exceeded) and the latter is given by a quantile for a particular probability level of interest (Bouallègue et al. (2015)). Since the output of the QR model are quantiles, it is reasonable to evaluate the performance with a
skill score which has been developed for predictive quantiles (Koenker and Machado (1999); Friederichs and Hense (2007)), known as the Quantile Score (QS). It is based on an asymmetric piecewise linear function, the so called check function, $\rho_\tau(y_i - q_{\tau,i})$, which is a function of the probability level $\tau$ ($0 < \tau < 1$) and the error between the observation $y_i$ and the quantile forecast $q_{\tau,i}$ for $i = 1, \ldots, N$, where $N$ is the sample size. The check function is defined as:

$$\rho_\tau(y_i - q_{\tau,i}) = \begin{cases} \tau(y_i - q_{\tau,i}) & \forall y_i \geq q_{\tau,i} \\ (\tau - 1)(y_i - q_{\tau,i}) & \forall y_i < q_{\tau,i} \end{cases} \tag{6}$$

and the QS results as the mean of the check function with penalties $1 - \tau$ and $\tau$ for under- and over-forecasting (see Bouallègue et al. (2015)):

$$QS = \frac{1-\tau}{N} \sum_{i:y_i < q_{\tau,i}} (q_{\tau,i} - y_i) + \frac{\tau}{N} \sum_{i:y_i \geq q_{\tau,i}} (y_i - q_{\tau,i}) \tag{7}$$

The CRPS compares the forecast probability distribution with the observation and both are represented as cdfs. If $F$ is the predictive cdf and $y$ is the verifying observation, Gneiting and Ranjan (2011) showed that the CRPS can be defined equivalently
as standard form,

$$CRPS(F,y) \quad = \quad \int_{-\infty}^{\infty} (F(t) - I\{y \leq t\})^2 \, dt, \quad \text{and as} \tag{8}$$

$$= \quad 2 \int_0^1 \left( I\{y < F^{-1}(\tau)\} - \tau \right) \left( F^{-1}(\tau) - y \right) d\tau \tag{9}$$

Thus, in the standard form (Equ. 8) an ensemble of predictions can be converted into a piecewise constant cdf with jumps at the different models (ensemble members), and $I\{.\}$ is a Heaviside step function, with a single step from 0 to 1 at the observed
value of the variable. The equivalence of Equ. 8 to Equ. 9 was noted by Laio and Tamea (2007). For the quantile forecast $q_\tau = F^{-1}(\tau)$, the integrand in Equ. 9 equals the quantile score, i.e. the mean of the check function (Equ. 6). That means the CRPS corresponds to the integral of the QS over all thresholds, or likewise the integral of the QS over all probability levels (Laio and Tamea (2007) and Gneiting and Ranjan (2011)). Hence, the CRPS averages over the complete range of forecast thresholds and probability levels, whereas the QS looks at specific $\tau$-quantiles; thus, it is more efficient in revealing deficien-
cies in different parts of the distributions, especially with respect to the tails of the distribution. Both verification measures are negatively oriented, meaning the smaller the better.



## 3   Results

The different combination methods have been applied to the flood forecasting system for the river Sihl at the station Zurich (Switzerland), where two meteorological forecasts, the 16 member ensemble system COSMO-LEPS (Montani et al. (2011)) and the deterministic C7 system (produced at MeteoSwiss with ≈ 7 km resolution) are implemented (a detailed description can

be found in Addor et al. (2011); Ronco et al. (2015); Liechti et al. (2016)). Forecasts are available from 2010-02-24 to 2016-04-27 once a day with hourly time resolution, and have been post-processed in order to derive predictive distributions and quantile forecasts. Since the 16 ensemble members are exchangeable, the numbering of the ensemble members is independent between consecutive forecast days. They are aggregated to one predictive pdf, resp. quantile forecast using probability and quantile averaging methods. It should be stressed that this simple averaging step only makes sense in the case of exchangeable

EPS, like the ENS from ECMWF (Molteni et al. (1996)) and the therein nested high resolution COSMO-LEPS, whereas for non-exchangeable Ensemble systems the following combination methods could be applied directly. The post-processing of the C7, using a QR method in combination with Neural Networks (QRNN, Taylor (2000); Cannon (2011)), will result in one forecast of quantiles. Both quantile forecasts from the averaged COSMO-LEPS and from the C7 results in direct estimates of the inverse cumulative density function (i.e. the quantile function), which in turn allows the derivation of the predictive

uncertainty (see for example (Weerts et al., 2011; López López et al., 2014; Dogulu et al., 2015)). If the number of estimated quantiles within the domain $\{0 < \tau < 1\}$ is sufficiently large, the resulting distribution could be considered as continuous. In Quiñonero Candela et al. (2006) the cdf, respectively pdf is constructed by combining step-interpolation of probability densities for specified $\tau$-quantiles with exponential lower and upper tails, which will be called the empirical method (EMP). Alternatively the pdf can be constructed by monotone re-arranging of the $\tau$-quantiles and fitting a distribution to these quan-

tiles for each lead-time (e.g. the log-normal (LN) distribution). The advantage of the proposed quantile re-arranging and the distribution fitting is twofold and efficiently prevents known problems occurring with QR: firstly it eliminates the problem of crossing of different quantiles (i.e. the unrealistic, but possible outcome of the non-linear optimization problem yielding lower quantiles for higher stream-flow values Chernozhukov et al. (2010), e.g. the value of the 0.90 quantile is higher than the value of the 0.95 quantile) and secondly it permits the extrapolation to extremes not included in the training sample (Bowden et al.

25  (2012)).

Thus, in total there are 6 different forecasts available after post-processing, two based on the application of the QRNN method for the COSMO-LEPS with probability averaging (p.aver.), resp. quantile averaging (q.aver.), two post-processed C7 forecasts based on QRNN with the EMP and the LN aproach, and one forecast based on the waveVARX method. Additionally the raw COSMO-LEPS forecast will be included in the combination procedures as well.


The weighting parameters of the combination methods are estimated by applying a moving window with a size of 7 days (168 hours) for optimization. Different window sizes have been tested as well, but 7 days was chosen finally as a trade-off between computing time and efficiency. In Fig. 1 an example of the temporal evolution of the hourly weights for $\tau = 0.05$ and





a lead-time of 48 hours for the three combination methods is shown (left panels) and the averaged weights conditioned on $\tau$ (right panels).

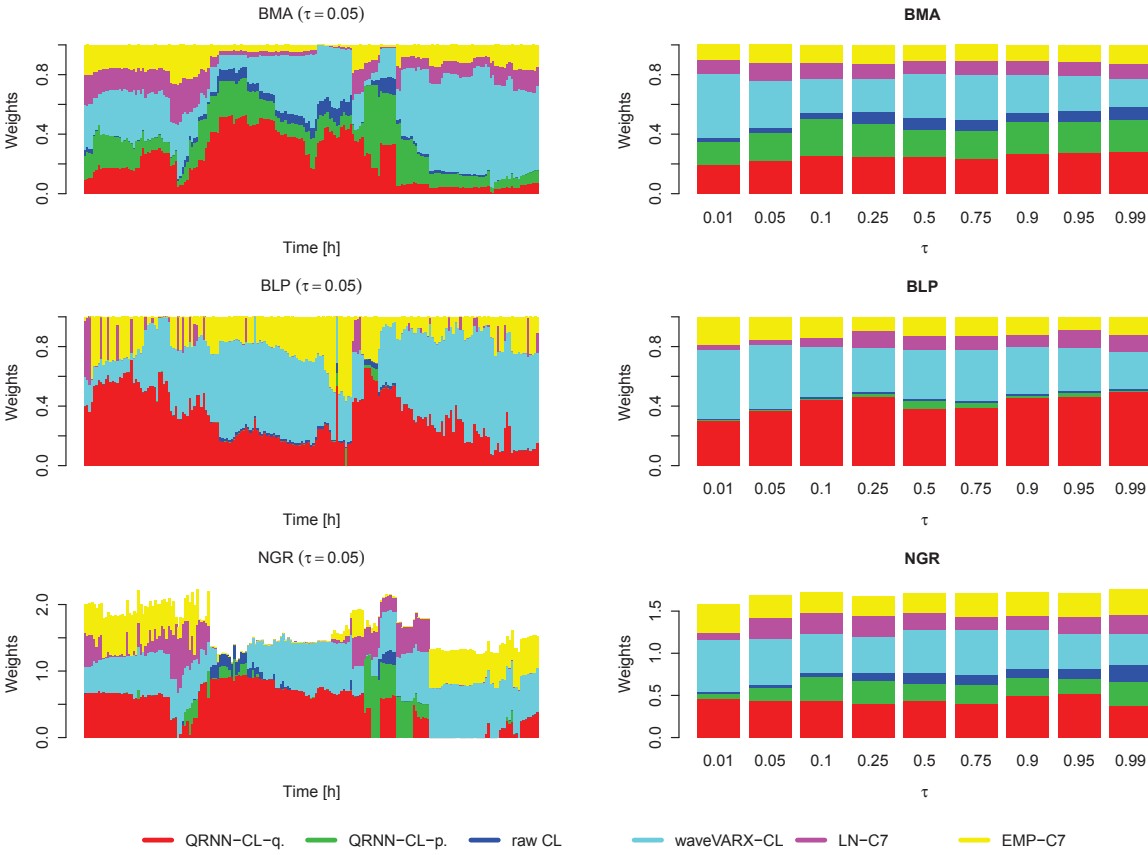

**Figure 1.** Hourly weights of the BMA (top row), NGR (middle), BLP (bottom row) method estimated for a lead-time of 48 hours and $\tau = 0.05$ (left column) and averaged weights conditioned on the probability level $\tau$ (right column). The 6 forecasts are the QRNN method for the COSMO-LEPS with probability averaging (QRNN-CL-p.), resp. quantile averaging (QRNN-CL-q.) and the waveVARX(-CL) method, two post-processed C7 forecasts based on QRNN with the EMP and the LN aproach and the raw COSMO-LEPS (CL) forecast.

In Figure 2 the CRPS results of these 6 forecasts are shown in comparison to the BLP in order to demonstrate the motivation of aggregating these systems. As can be seen clearly, the combined forecast outperforms each of the individual forecasts in view of the CRPS. The question now is whether there are significant differences between the three combination methods. Although modifications for the NGR exists for non-normal distributed variates (see for example Baran (2014), Baran and Lerch (2015)), for the application of the NGR and the BMA the stream-flow values have been transformed to the Normal Space by the help of the Normal Quantile Transformation (Van der Waerden (1952), Van der Waerden (1953a, b)).





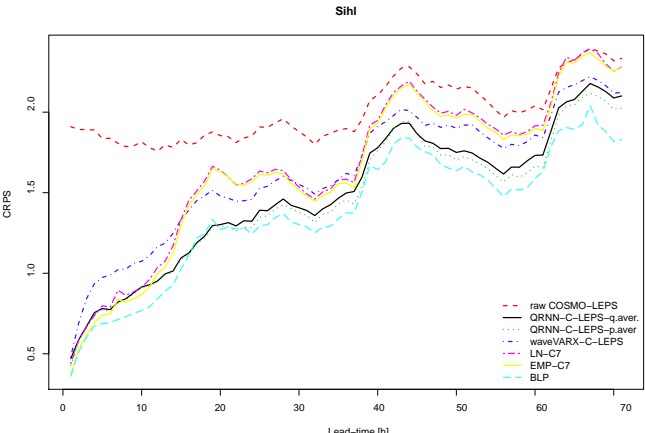

**Figure 2.** CRPS of the raw COSMO-LEPS (C-LEPS), the five post-processed forecasts and the BLP combined forecast

Before the forecast skill of the the three combination methods are compared, the statistical consistency between the predictive cdf and the observations are analysed with the help of the Probability Integral Transform (PIT) as proposed by Dawid (1984) (see Fig. 3). In case of well calibrated forecasts, the sequence of PIT values will follow a uniform distribution $U(0,1)$. U-shaped PIT histograms indicate underdispersed forecasts with too little spread on average, inverse U-shaped histograms correspond to overdispersed forecasts (see for example Gneiting et al. (2007), Laio and Tamea (2007)).

In Fig. 4 the results of the QS at four lead-times for the raw COSMO-LEPS (C-L, black line) and for the three combination methods BLP (red line), NGR (green line), BMA (blue line) are shown and compared to the QS results of the raw C-L (black circles). Additionally, a simple Quantile Mapping (QM) is applied (cyan diamonds) to the raw C-L forecasts in order to evaluate the positive effect of using more complex methods. Thereby the cdf of the raw forecast is matched to the cdf of the observations. As mentioned in Zhao et al. (2017) QM is highly effective for bias correction, but ensemble spread reliability problems cannot be solved properly.

The integral of the QS results in the CRPS, which is shown in Fig. 5 for the raw C-L, the QM approach and the three combination methods.

## 4 Discussion

So far most of the studies comparing the results of the BMA and the NGR approach did not find any preference (see for example Williams et al. (2014)). In this paper these two methods are checked against the BLP, which has not been used for hydrological purposes until now. In a first step the weights derived for each individual, raw and post-processed, forecast system are compared. The pattern of these optimized weights in Fig. 1 (left column) exemplified for a short time period show rather vague similarities between the three combination methods. The BLP and the NGR are in general more spiky with rapid changes between consecutive hours. This could result from problems on convergence from the optimization algorithm applied





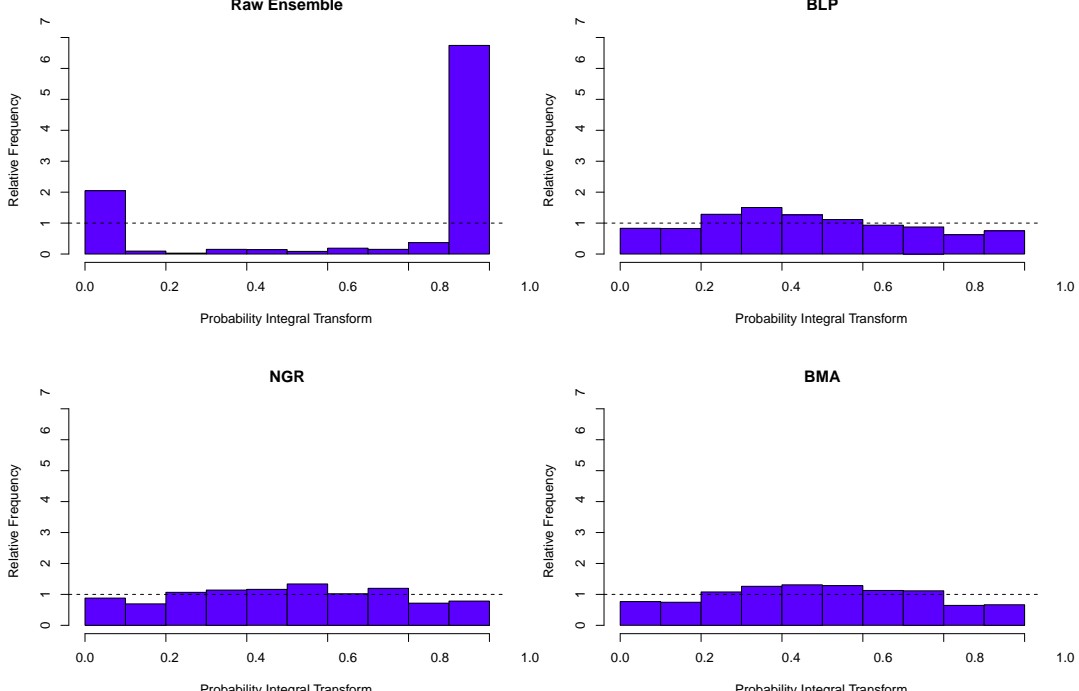

**Figure 3.** Probability Integral Transform (PIT) of the raw and the three combined forecast at a lead-time of 24 hours

for estimating the parameters ("constrOptim" in R (R Core Team (2015))).

The weights averaged over the the whole forecast period and separated in 9 stream-flow classes, i.e. probability levels $\tau$ show some slight tendencies, namely that the waveVARX method is better suited for low flow conditions, whereas the importance of the QRNN approach increases with $\tau$ for the Sihl catchment.

5     The results of the PIT clearly indicate that all three combination result in well-calibrated forecasts with close to uniform histograms. In Fig. 3 the examples for the 24h forecast are given, highlighting the heavy underdispersiveness of the raw forecasts. The same behaviour is visible for almost all lead-times, however the raw COSMO-LEPS forecasts are getting less underdisperse with increasing lead-time, since the spread and the uncertainty in the ensemble increases.

10     The analysis of the QS (Fig. 4) show slightly better results for the BLP followed by the NGR and BMA. The raw COSMO-LEPS (C-L) and the QM are much worse, especially for smaller lead-times. It is interesting to see that the QS of the raw C-L follows a straight line for smaller lead-times (6 and 12 hours) in the same manner as one would expect from deterministic forecasts, because of the under-dispersiveness of the C-L at the beginning of the forecast horizon. The slope of this line is an indicator of the size of the (positive) bias. The QM at a lead-time of 6 hours is also a straight line, however with an opposite, but

15    much smaller and negative slope (bias) in comparison to the raw C-L. With increasing lead-times the QS of the raw C-L and the QM forecasts come closer to the combined forecasts for probability levels between 0.1 and 0.5. This is caused by the increased

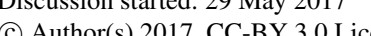



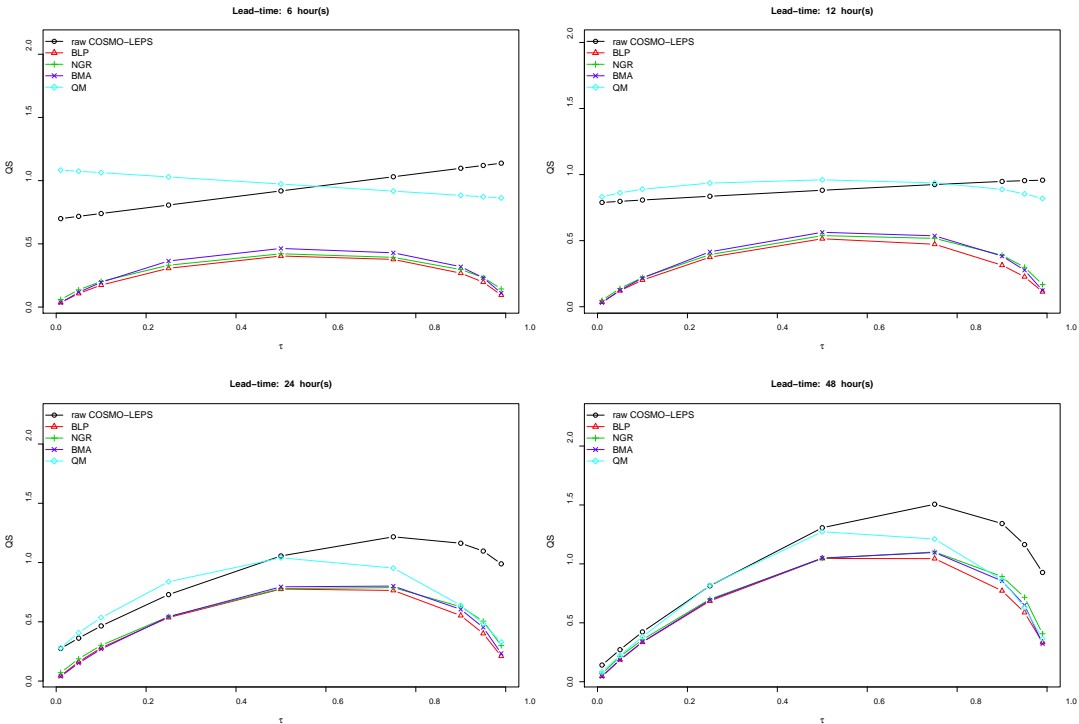

**Figure 4.** Quantile Score (QS) for various lead-times and the three combination methods in comparison to the raw COMSO-LEPS and a simple Quantile Mapping (QM) approach

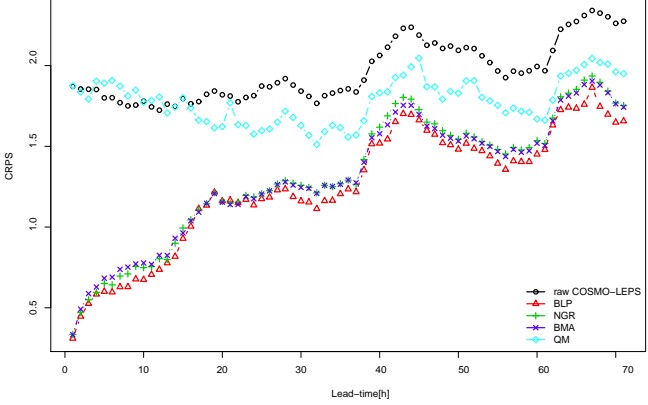

**Figure 5.** CRPS of the raw and the combined forecast



spread of the ensemble. However, for a lead-time of 24 and 48 hours the raw C-L forecasts still show the worst behaviour at higher flows, whereas the QM method performs at a lead-time of 48 hour almost as well as the combination methods, apart from the forecasts around the median.

As already stated previously, the comparison of the CRPS of the different post-processed methods and the aggregated ones (e.g. BLP) clearly identifies the advantage of combination (Fig. 2). The CRPS for the different combination methods (Fig. 5) confirm the results of the QS. In general the results of the BLP are slightly better than the NGR and the BMA results. It seems that for those periods of lead-times, where the BLP is not superior (e.g. around 20 hours), the optimization routines had problems on convergence. However further analysis will be necessary. The comparison with the QM approach confirmed the

results of Zhao et al. (2017), since the forecast quality did not show any improvements at the first lead-times because of the underdispersiveness of the raw C-L. Thus, the more complex combination by far outperforms the QM method.

## 5   Conclusions

Combination is an essential tool for improving the forecast quality. The different methods are all more or less equally suited. Although the BLP showed slightly better results, the straight forward application and the low computational costs of the NGR

make this method an equally good alternative, at least for this case study. The parameter estimation of the BMA and the BLP could get quite time consuming and sometimes results in suboptimal solutions, which could degrade the gain of applying combination methods.

*Competing interests.* The authors declare that no competing interests are present

*Acknowledgements.* The real-time operational system for the Sihl basin is financed by the Office of Waste, Water, Energy and Air of the

Canton of Zurich. This study was conducted in the framework of the Swiss Competence Center for Energy Research - Supply of Electricity (SCCER-SoE) with funding from the Commission for Technology and Innovation CTI (grant 2013.0288). MeteoSwiss is greatly acknowledged for providing all used meteorological data. The Swiss Federal Office for Environment (FOEN) provided the observed discharge data. The authors would like to thank especially Vanessa Round for proofreading.





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
