# Peer review of "Technical Note: Combining Quantile Forecasts and Predictive Distributions of Stream-flows"

_Hydrology and Earth System Sciences, 2017_

## Referee Comment (RC1) · Anonymous Referee #1 · 9 Jun 2017

Forecast combination is an important and topical problem. In this light, the paper is a welcome hydrologic case study.

However, I was unable to appreciate the technical details, and couldn't follow the comparison between Bayesian Model Averaging (BMA), Nonhomogeneous Gaussian Regression (NGR/EMOS) and the Beta transformed Linear Pool (BLP), for the following reasons.

1. Combination methods

BMA and NGR/EMOS convert a set of M point forecasts, say $k_1$, ..., $k_M$ into a single, combined predictive distribution. In the case of BMA, the combined predictive

distribution is a mixture distribution, in the case of NGR/EMOS it is a single Gaussian distribution.

In contrast, BLP converts a set of predictive distributions, say $F\_1, ..., F\_M$, into a single, combined predictive distribution.

As I understand Section 3, the authors produce a set of M = 7 predictive distributions (six postprocessed ones, and the raw COSMO-LEPS ensemble), say $F\_1, ..., F\_7$. This clearly fits the BLP framework, except that I do not understand how the quantile forecasts were converted into predictive distributions. How was this done?

And how was BMA and NGR/EMOS applied? As noted, BMA and NGR/EMOS require point forecasts as input. So did you reduce the predictive distributions $F\_1, .., F\_7$ to point forecasts, e.g., by computing their respective means $k\_1, ..., k\_7$, before applying BMA and NGR/EMOS? Is this what you did? If so, how did you proceed in the quantile case? If not, what else was done?

How are the BMA kernel distributions specified? Which distribution family do they belong to? Normal, Gamma?

2. Combination weights

Figure 1 shows "[h]ourly weights" for BMA, BLP and NGR. On the left-hand side, what period of time is represented by the horizontal axis? On the right-hand side, combination weights are plotted in their dependence on the "probability level". This I don't understand; perhaps "quantile level" is meant, but even then neither BMA nor NGR/EMOS nor BLP have weights that vary with a quantile level. What is shown here?

3. Suggested reorganization

Moving (thoroughly edited and expanded versions of) the first two paragraphs in the "Results" section to the beginning of the "Methods" section would make the "Methods" section, and the paper as a whole, easier to understand. Please use the set of predicitive distributions $F\_1, ..., F\_7$ as a starting point, and then explain how these

distributions serve as inputs to BMA, NGR/EMOS, and BLP.

---

## Author Comment (AC1) · 19 Jun 2017

The authors want to thank Referee #1 for the very helpful comments and we will try to incorporate them in an updated version in more detail! Beforehand we want to answer the questions in order to give some clarifications and missing essential information.

Combination methods

How to convert quantile forecasts into predictive distributions? On page 6, line 16 - 20 it is outlined that there have been two different approaches tested: a. Fitting a lognormal distribution to the quantiles (Bogner, et al ., 2016) b. Linear interpolation

and exponential tails (Candela, 2006) We will try to include more information about these proposed methods.

Regarding the BMA and the NGR approach: for all cases (based on quantiles and on predictive distributions) the 0.5 quantiles have been used as a point forecast for BMA and NGR. In case of quantile forecasts the 0.5 quantile has been estimated from the lognormal approximations – see above.

The Normal distribution has been used as a kernel for the BMA. Thus, all the data (observed and forecast) have been transformed into the Gaussian space before applying the BMA and the NGR (see also page 7 line 5-8).

Combination weights

Fig. 1 covers the period from 2010/08 – 2016/04 with hourly resolution – we will add the time information on the x-axis on the left side. Whereas on the left the example of the \tau=0.05 is shown, on the right the 9 different \tau's are shown, thus the second bar corresponds to the left panel averaged over the whole time period.

Suggested reorganization

We will re-write the paper according to the suggested reorganization.

---

## Referee Comment (RC2) · Anonymous Referee #2 · 30 Jun 2017

The study provides a comparison of different methods for combining probabilistic forecasts of streamflows. It finds that the methods included in the study performed more or less the same, all led to improvement over the raw ensembles. The study is of interests to the ensemble hydrological forecasting community and has good material to present.

I believe the writing of the paper needs to be substantially improved before it is suitable for publications. I found it frustrating to read, as information is not always complete or logically organised. For example, • I cannot make sense of the section before 2.1 under Methods; • The first paragraph under Results is a way too long to read; • Following Figure 2 on CRPS for raw forecasts and comparison with BLP forecasts, it

would be logical to show Figure 4 for comparison of CRPS from different aggregation methods, rather than going to PIT histogram (Figure 3) first; • In Figure 4, it will be good to denote COSMO-LEPS raw forecasts (as in other figures), rather than just raw forecasts; • In the first paragraph under Results, it states: "Since the 16 ensemble members are exchangeable, the numbering of the ensemble members is independent between consecutive forecast days". This seems to me a bit strange.

The above list is by no means exhaustive. There is room to sharpen up the writing throughout the paper.

With effort from the authors to improve the writing, the paper will be a good one to publish.

---

## Author Comment (AC2) · 4 Jul 2017

The authors want to thank Referee #2 for the helpful comments and we will try to reorganize the paper accordingly, sharpen the writing and incorporate all the comments in an updated version!

---

## Author Response (AR1)

The authors want to thank both reviewers for their very helpful comments and we have tried to incorporate the requested changes accordingly.

Reviewer 1:

Forecast combination is an important and topical problem. In this light, the paper is a welcome hydrologic case study.
However, I was unable to appreciate the technical details, and couldn't follow the comparison between Bayesian Model Averaging (BMA), Nonhomogeneous Gaussian Regression (NGR/EMOS) and the Beta transformed Linear Pool (BLP), for the following reasons.
1. Combination methods
BMA and NGR/EMOS convert a set of M point forecasts, say k_1, ..., k_M into a single, combined predictive distribution. In the case of BMA, the combined predictive distribution is a mixture distribution, in the case of NGR/EMOS it is a single Gaussian distribution.
In contrast, BLP converts a set of predictive distributions, say F_1, ..., F_M, into a single, combined predictive distribution.
As I understand Section 3, the authors produce a set of M = 7 predictive distributions (six postprocessed ones, and the raw COSMO-LEPS ensemble), say F_1, ..., F_7.
This clearly fits the BLP framework, except that I do not understand how the quantile forecasts were converted into predictive distributions. How was this done?

Response:

We have included some more details about the conversion between quantile forecasts and predictive distributions in the Methods section (page 3, line 2 – 15), where we outline that we have tested two different approaches: (a) Linear interpolation and exponential tails (Candela, 2006) and (b) Fitting a lognormal distribution to the quantiles (Bogner, et al ., 2016)

And how was BMA and NGR/EMOS applied? As noted, BMA and NGR/EMOS require point forecasts as input. So did you reduce the predictive distributions F_1, .., F_7 to point forecasts, e.g., by computing their respective means k_1, ..., k_7, before applying BMA and NGR/EMOS? Is this what you did? If so, how did you proceed in the quantile case? If not, what else was done?

Thank you for highlighting this lack of information, which is quite essential and has been included now (page 4, line 11; page 5 line 13)

For all cases (quantile forecasts and predictive distributions) the 0.5 quantiles have been used as a point forecast for BMA and NGR.

How are the BMA kernel distributions specified? Which distribution family do they belong to? Normal, Gamma?

The Normal distribution has been used as a kernel for the BMA. Thus, all the data (observed and forecast) have been transformed into the Gaussian space before applying the BMA and the NGR (see also page 3 line 28-30, page 4 line 8-9, page 5 line 13-14).

2. Combination weights
Figure 1 shows "[h]ourly weights" for BMA, BLP and NGR. On the left-hand side, what period of time is represented by the horizontal axis? On the right-hand side, combination

weights are plotted in their dependence on the "probability level". This I don't understand; perhaps "quantile level" is meant, but even then neither BMA nor NGR/EMOS nor BLP have weights that vary with a quantile level. What is shown here?

We have changed Fig. 1, which is Fig. 2 now, and have included the information about time (x-axis). In the previous version we have split the observations into intervals representing the 9 quantiles and assigned the weights according to these intervals. We though it could be interesting to see the differences of the weights conditioned on quantiles, but maybe that Figure was not too informative, so we have excluded it.

3. Suggested reorganization
Moving (thoroughly edited and expanded versions of) the first two paragraphs in the "Results" section to the beginning of the "Methods" section would make the "Methods" section, and the paper as a whole, easier to understand. Please use the set of predictive distributions F_1, ..., F_7 as a starting point, and then explain how these distributions serve as inputs to BMA, NGR/EMOS, and BLP.

Thank you for this suggestions. We totally agree that this reorganization will help to make this paper easier to read. Therefore we have also included  Fig. 1, which gives an overview of the different forecast models used as input for the combination methods.

Reviewer 2:

According to your suggestion we have  reorganized the paper and moved parts from the Results section to the Methods section.  Also we have reorganized the structure of the figures, the results and the discussion section according to your suggestion. The sentence regarding the exchangable members of COSMO-LEPS has been removed.

Once again we want to thank both reviewers for their time reviewing the paper and their comments!

[revised manuscript text omitted]

---

## Author Response (AR2)

The authors want to thank once again the reviewers for their very helpful comments and their suggestions to improve the readability of the paper!!

Referee #2:

1. The authors should acknowledge that there are combination methods that use the whole forecast distributions directly, rather than taking the means or medians of the forecast distributions only. Examples are Wang et al (2012) and Schepen and Wang (2015).

We want to thank you for pointing out the missing the references, which have been included now:

Page 4, line 9-14:

In the work of Wang et al. (2012) and Schepen and Wang (2015) variants of the BMA method have been applied, which allow the direct usage of the cdf's for estimating the weighting parameters. However, in this study these BMA approaches have not been implemented and the estimated medians ($\tau = 0.5$) from the five post-processing methods and from the raw COSMO-LEPS are taken as input only in order to allow better comparison with the following NGR approach.

2. The authors need to clarify whether steps were taken in the processing of the forecasts before the combination to ensure independence between model fitting and later verification (such as by cross validation, split sample, or fitting to only data before event being evaluated.

We have included now the following sentences:

Page 7, line 18-23:

[revised manuscript text omitted]